# Behavioural activation activities for employees in the Chinese culture: A workshop

Sijin Sun[1]*, Yao Xiao[2], Zheyuan Zhang[1], Celine Mougenot[1], Nick Glozier[3], Rafael A. Calvo[1]

1 Imperial College London, London, United Kingdom, 2 Independent Psychologist, London, United Kingdom, 3 The University of Sydney, Sydney, Australia

* s.sun20@imperial.ac.uk

**Data Availability Statement:** All relevant data are within the paper and its Supporting Information files.

**Funding:** The authors received no specific funding for this work.

## Abstract

Behavioral Activation Therapy (BAT) is widely used in Western countries, and digital interventions based on BAT are also increasingly common. This study explored how BAT can be adapted for Chinese employees. Through twelve co-design workshops conducted online, a total of 46 Chinese employees actively participated in the process of defining positive activities for behavioural activation therapy. Using Hofstede's cultural dimensional theory as a framework and considering the traditional influence of Confucianism and the dynamic nature of China's contemporary socio-cultural transformation, we identified and examined culturally sensitive and controversial activities that emerged during the study. Our findings indicate that Chinese participants, when compared to their western counterparts, generally displayed less favourable attitudes towards activities such as extreme sports, religion, charitable work, family or after-work social activities. Additionally, they demonstrated less willingness to express emotions openly, provide constructive upward feedback, or seek assistance from mental health experts. Our research indicates that the implementation of behavioural activation activities, as validated in Western literature and classified into three layers in our study—general, workplace-related, and mental health-related—needs to be carefully adapted to align with the current Chinese cultural and societal context, this includes recommending leisure activities that are lighter and less risky, provide cultural sensitive advisory that facilitate effective workplace communication, and consider offering mental health self-help knowledge toolkits.

## 1. Introduction

### 1.1 Behavioural activation therapy

Behavioural activation is a structured therapy based on behaviourism. By monitoring activities and formulating strategies, this therapy encourages participants to engage in healthier and mood-enhancing behaviours, thereby enhancing their cognition, emotions, and overall wellbeing [1]. Behavioural activation-based interventions have proven to be effective in Western

**Competing interests:** The authors have declared that no competing interests exist.

populations and have been widely documented in the literature for addressing various mental health issues such as depression [2] and loneliness [3].

While existing literature on behavioural activation predominantly focuses on Western populations, behavioural activation therapy has also been studied in Eastern populations, including China, and across age groups. In a counselling case study [4], behavioural activation therapy targeted Chinese college students with social anxiety disorder and resulted in a significant improvement in self-reported anxiety and depression scores following a five-stage counselling treatment. Xie et.al. [5] conducted a randomised controlled trial on 80 elderly "left behind" Chinese farmers who were over 60 years old and lived in their hometown with all their children away [6], showing that behavioural activation therapy can ameliorate depressive symptoms in this population.

Apart from traditional clinical treatment, recent studies have integrated behavioural activation-based interventions with digital support technology. A randomised controlled trial involving 2,275 working Australians [7] demonstrated the potential benefits of behavioural activation that encouraged participants to engage in "positive activities" and was delivered through a smartphone. However, few studies have explored the Asian context and what "positive activities" means there or how to methodologically consider such cross-cultural perspectives. Consequently, a research gap exists, particularly in culturally sensitive design of digital mental health support technology within the Chinese context [8].

As one of the disciplines for designing behavioural activation therapy is that the positive activities need to be those "associated with the experience of pleasure or mastery" [9], the perspectives and feelings of target users need to be understood and respected. Prior studies [10, 11], have indicated that psychotherapies such as Cognitive Behavioural Therapy and Behavioural Activation yield greater efficacy when appropriately culturally adapted. Therefore, it is imperative to understand which activities are deemed positive and appropriate within the users' culture.

## 1.2 Cultural dimensions theory and Chinese socio-cultural context

Cultural dimension theory was developed by Geert Hofstede [12, 13] and is widely used to understand cultural differences in the workplace. Through conducting surveys with thousands of IBM employees in 53 countries around the world, he proposed different cultures are varied in six main dimensions, including power distance, individualism-collectivism, masculinity-femininity, uncertainty avoidance, long-term orientation, indulgence-restraint. Amongst the 53 countries analysed, China demonstrates relatively high scores in long-term orientation, power distance, masculinity, and indulgence. However, it receives lower scores in individualism and uncertainty avoidance. In contrast, Australia, where previous research has utilised digital support technology to enhance behavioural activation therapy, exhibits significantly lower power distance and long-term orientation. Additionally, Australia showcases higher levels of individualism, uncertainty avoidance, and indulgence compared to China.

When culturally adapting interventions to China its important to consider Confucianism which has influenced China since the Han dynasty (206 B.C.E. to 220 C.E.), setting moral obligations and hierarchical rules for maintaining group harmony as core social principles [14]. Nevertheless, despite the impact of the traditional Confucianism value, China has undergone profound transformations specially since 1978 when the Chinese government opened its doors to Western trade and investment. China has turned into a market economy and big cities are becoming more westernised, both economically and culturally. These muti-facet layers of China's cultural context are essential when designing behavioural interventions [15].

Chinese employees are currently facing significant mental health challenges. According to the *2022 Report on Workplace Employees' Psychological State* [15], over half (55.7%) of the

respondents were found to be in a state of "suboptimal mental health," with 6.4% experiencing severe emotional stress and psychological disorders.

This study builds on research on a mental health intervention for Australian employees in male dominated industries [7], aiming to improve the effectiveness of behavioural activation specifically in the Chinese context. It investigates the perspectives of Chinese employees toward positive activities as understood in their culture. We will also use the culture dimensional theory to further analyse the results of in a culture specific context.

## 2. Methods

### 2.1 Overview

The primary objective of this study was to employ a participatory co-design approach, aligned with STROBE guidelines (S1 Checklist), to develop and refine positive activities suitable for behavioral activation therapy (BAT) within the Chinese cultural context.

All experimental protocols were approved by the Imperial College Research Ethics Committee (ICREC) of Imperial college London (project 22IC7794). The study was conducted in accordance with established ethical guidelines and principles, ensuring the protection of participants' rights, privacy, and confidentiality. All participants were fully informed about the procedure and aims of this study in simplified Chinese. Written informed consent forms were obtained before the study from each participant.

Twelve co-design workshops were conducted online, involving a total of 46 Chinese employees. Eligible participants were required to be Chinese nationals, currently employed or previously employed, and have access to computers and the internet. The mental well-being levels of the participants were not assessed and did not serve as a criterion for their selection. Participants were introduced to the concept of BAT intervention and were presented with 201 distinct positive activities for behavioural activation, originally designed for a digital BAT intervention for Australian employees by Australian psychologists. In each workshop group, participants were told to vote "likes" and "dislikes" for the activities on a collaborative online whiteboard and they were encouraged to discuss with each other around these BAT activities. This mixed-methods study combined quantitative methods, where participants' votes on each activity on the Miro board were recorded and statistically analyzed, with qualitative methods, capturing and thematically analyzing conversation content from their discussions. These methods were chosen because they facilitate an inclusive, collaborative environment that harnesses the diverse perspectives of participants, essential for culturally sensitive adaptations.

The study results were generated from the workshop as in the following flowchart of Fig 1.

### 2.2. Workshop

The workshops were facilitated by a Chinese researcher, and local technical support was available in two instances to assist participants with limited computer skills. The number of Chinese employees participating in each workshop varied between 2 and 6, and the duration of the workshops ranged from 1.5 to 3 hours. The online collaborative whiteboard tool, "Miro," was utilized to facilitate the workshops.

The workshop sessions were structured into three parts: (1) Introduction, (2) Discussion and Design of BAT Positive Activities, and (3) General Discussion on Preferences for Digital BAT Support.

**2.2.1 Participants.** The study included a total of 46 participants (47.8% male) who actively participated in 12 workshops. Participant recruitment was advertised on Mandarin Chinese social messaging platforms as posts, encouraging people with interests to contact the researchers or repost to potential subjects. No financial compensation was given. Participants were

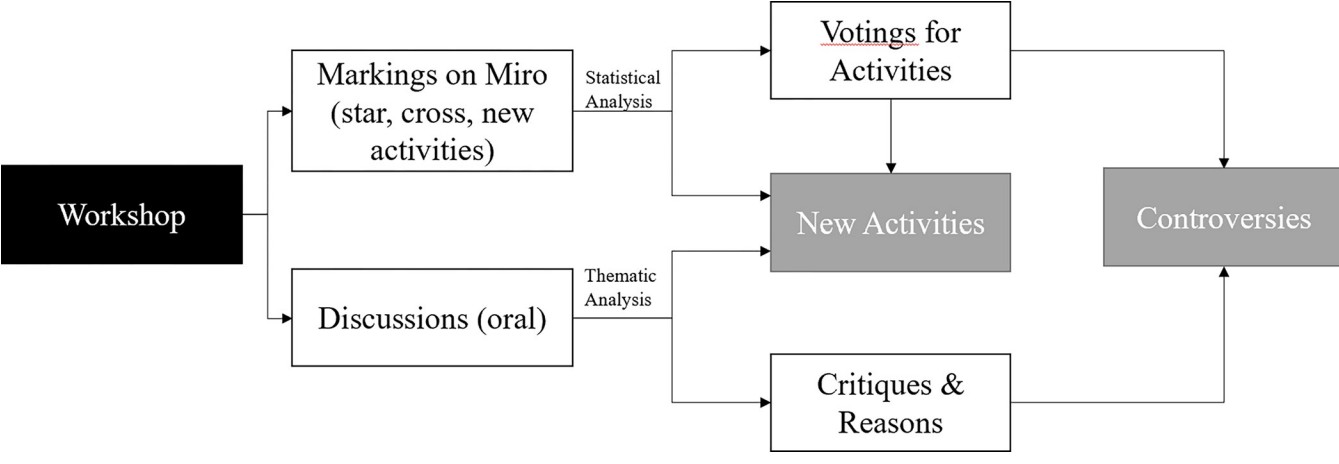

**Fig 1. Flowchart.**

recruited online from 1st September 2022 to 15th September 2022. The dissemination of the contact link was stopped after 50 participants were recruited, whose working experience ranged from 1 year to 30 years, reflecting a diverse range of professional backgrounds. Throughout the research, 4 (8%) of the 50 recruited participants dropped out before the workshop started. Another two (4%) participants left the workshop midway due to urgent work obligations, with their statements prior to leave were still recorded for qualitative analysis with their consent. The rest 44 (88%) of them completed the workshop tasks to the end.

Throughonline participant information sheets, all participants were fully informed about the procedure and aims of this study in simplified Chinese. Online consent forms were obtained before the study from each participant. They were allocated into 12 workshops based on their time preference and educational levels to ensure they can understand each other effectively. Each participant only participated in one workshop.

**2.2.2 Procedure.** Initially, participants were asked to complete an anonymous online form providing their demographic information. They then introduced themselves using nicknames and shared their working backgrounds in a brief introduction session. Participants were advised not to disclose any personal details, including their real names, during this session. The aim of the workshop was explained and participants were introduced to the concept of BAT. Participants were then familiarized with the basic features of the Miro tool. Each participant was requested to choose a unique colour to represent themselves for subsequent interactions on the whiteboard.

On the shared online whiteboard, participants were presented with a list of 201 distinct positive activities originally co-designed for the Australian digital BAT intervention Headgear by Australian psychologists and employees. These activities (as listed in S1 File) were categorized into five major themes: recreation, relationships, body and mind, work and study, and daily life. Each theme comprised 3 to 7 subcategories further classifying the activities.

Participants were encouraged to assess and evaluate the appropriateness of these activities for Chinese employees. As demonstrated in Fig 2, if an activity was deemed valuable, feasible, and positive, participants marked it with a "star" symbol using their assigned colour. Conversely, if an activity was considered worthless, impractical, or negative, participants marked it with a corresponding "cross" symbol. Participants were also encouraged to contribute new activity ideas suitable for the Chinese context by adding a new sticker in their assigned colour and documenting the idea. Furthermore, participants were actively encouraged to engage in discussions with other group members to share their thoughts and perspectives on the

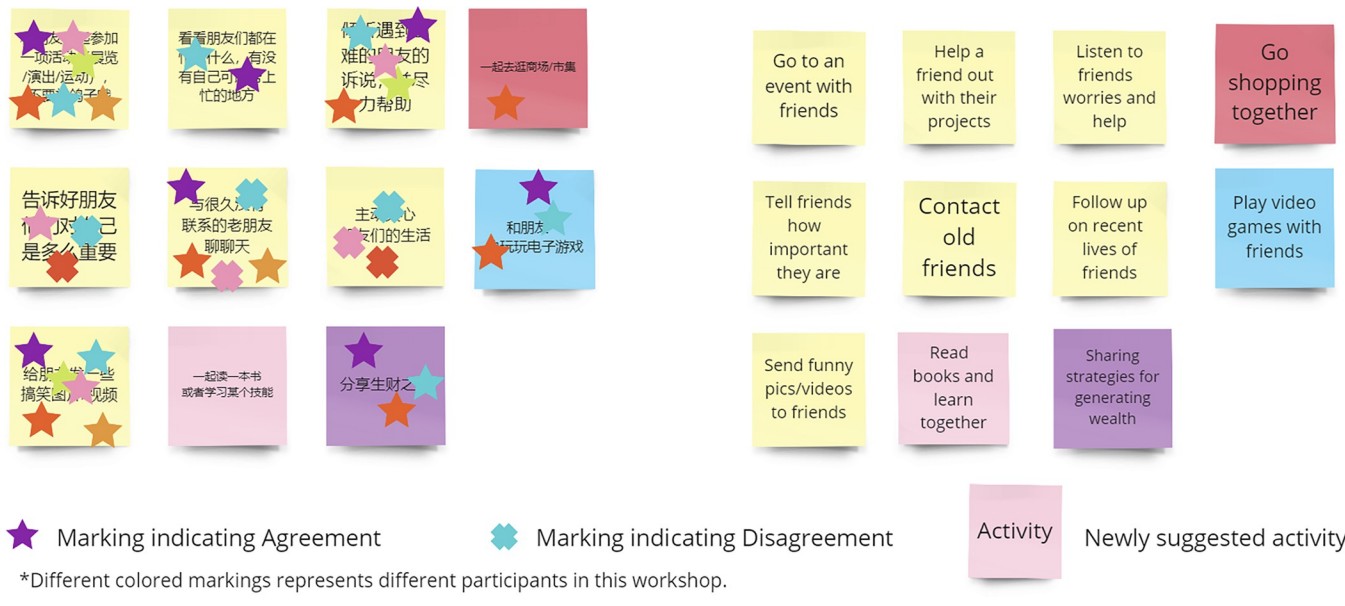

Fig 2. An example of markings on activities.

activities. The researcher minimized interference in the discussions among participants and allowed ample time for engagement, resulting in varying workshop durations ranging from 1.5 to 3 hours.

## 2.3 Data collection and data analysis

**2.3.1 Data collection.**   The "star" and "cross" markings on the online whiteboard provided by the participants were recorded and subjected to statistical analysis with respect to participants' gender, working history, educational level, and experience of living abroad. (The inclusion of living abroad as a variable was justified by the hypothesis that participants' exposure to Western cultures could potentially explain their understanding of and reactions to the Western cultural influences behind the design of the original activities.) Furthermore, all new activities proposed by the participants on the whiteboard were recorded (as provided in S2 and S3 Files), and later carefully examined, synthesized, and categorized according to their corresponding themes during the analysis. The discussion audios of the workshops were recorded and transcribed by a bilingual researcher after the workshop.

**2.3.2 Data analysis.**   The markings of the 12 workshops were aggregated to identify the controversial activities. NE scores and DF scores were calculated for each activity. The NE score = number of "cross" markings / number of participants who are presented with the activity, indicates the percentage of participants marking "cross" (meaning that employees have negative views about the activity). The DF score = (number of "star"–number of "cross") / number of participants who are presented with the activity, indicates the percentage difference of participants holding positive and negative views on these activities (a low score represents notable internal controversies among participants).

An inductive thematic analysis [16] of the transcribed text was conducted by a bilingual designer with psychology background and a bilingual psychologist using NVivo (a qualitative data analysis software developed by QSR International). By applying thematic analysis to the conversations, controversial topics emerged, including workplace stress, helping others, charitable acts, and professional mental health support.

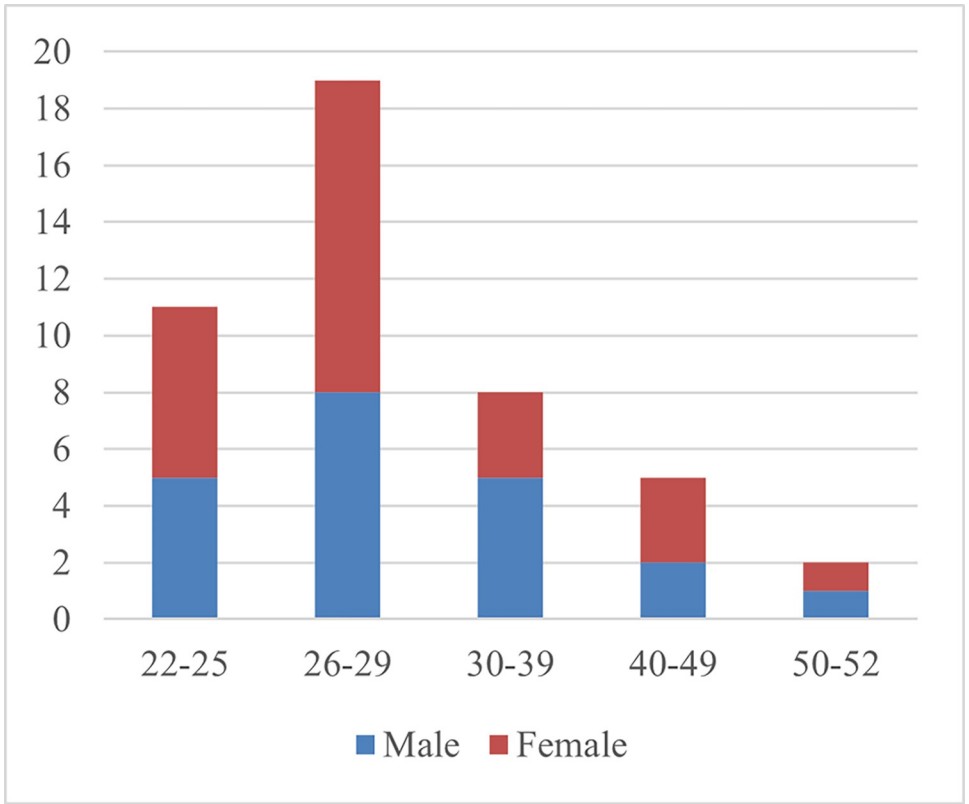

**Fig 3. Age distribution.**

## 3. Results

### 3.1 Demographics

The age of the participants in the study ranged from 22 to 52 (Fig 3), with 22 (48%) of the 46 being male. The educational levels of the participants varied from middle school to doctoral (Fig 4), indicating a broad spectrum of educational attainment within the sample. Of the 46 participants, 12 (26%) have had experience living abroad for more than two years.

### 3.2 General controversial activities

The research focused on identifying and examining the "Controversial Activities" that emerged during the analytical process. These"Controversial Activities" were observed at two distinct layers. Firstly, the results conveyed culture-specific messages that diverged from the corresponding findings in Australia [17], so we will understand the cross-culture differences for the definition of "positive activities.".Secondly, the results garnered diverse opinions within the Chinese participant group, therefore we can understand the within group differences in Chinese participants towards the"Controversial Activities"The subsequent section will delve into these activities qualified as "Controversial Activities" in detail.

Many activities discussed during the workshop sparked debates among the participants. This section will focus on non-work-related activities that generated differences either in comparison to the original studies conducted in Australia [17] or within the Chinese participant group.

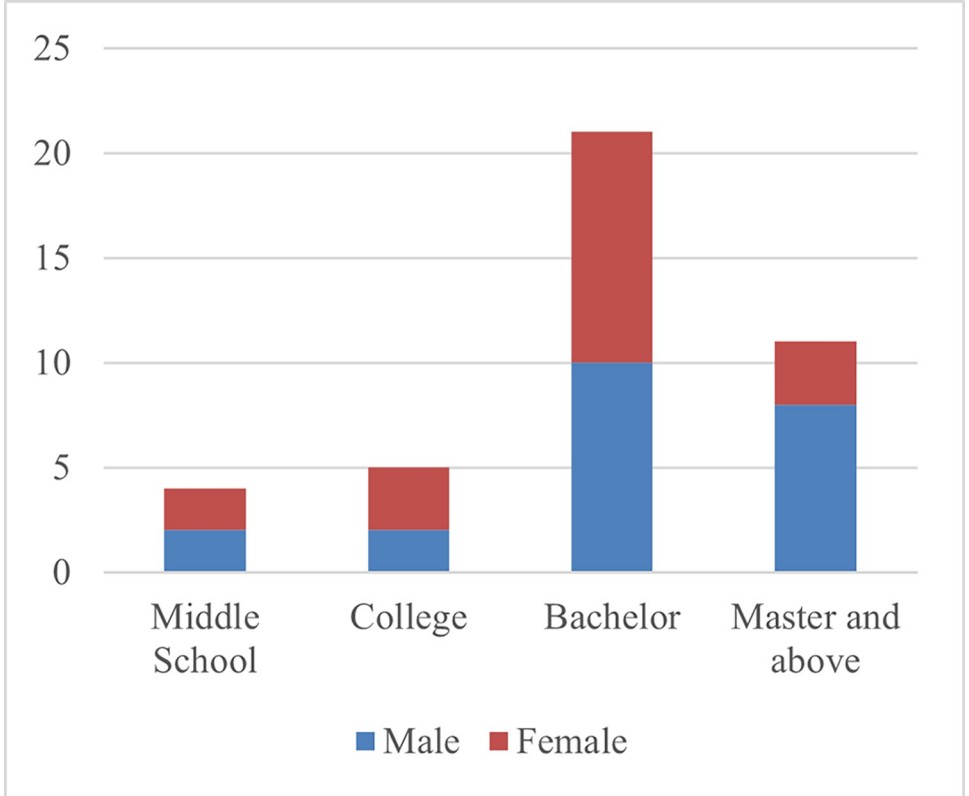

**Fig 4. Educational level distribution.**

### 3.2.1 Sports

*Why swim before work when I am already so tired? To torture myself? ["Purrloin", Male, 27, Construction Worker]*

Sports-related activities were a subject of intense discussion among the participants. One activity that garnered significant debate was "Wake up an hour early and go for a swim before work," Chinese participants expressed scepticism, considering it "unrealistic." Also, Participants argued that exercising under such circumstances would be "torturous" due to exhaustion and the subsequent difficulty in achieving relaxation. Furthermore, the limited availability of early morning swimming pools posed another obstacle. Despite these factors, Chinese participants generally exhibited a positive attitude towards swimming itself. Similarly, "Water sports" and "joining sports teams in your area" were also deemed inaccessible by many Chinese participants. "Going for a trial session at the gym or fitness community" received criticism due to the prevalence of so-called "free" activities that are, in reality, promotional offers aimed at enticing individuals into long-term membership commitments.

**3.2.2 Financial habits.** Participants engaged in discussions around the activity of "Start setting aside $10 per week to put towards a monthly sporting or cultural event." This activity was met with opposition from 30.4% of the participants, while only 6.5% expressed support. Many Chinese participants turns out struggled to connect with the idea of saving money specifically for particular events and questioned the need for such an approach. As one participant

aptly expressed, "Does there have to be a cause for Western people to save money?" *["Magi-karp", Female, 26, Biomedical Researcher]*

**3.2.3 Charitable act.** Unexpectedly, many activities under the category of "helping others" received negative comments, including "Become a community volunteer," "Provide food for the homeless," and "Participate in charitable events." participants argue that it would be more effective to contact the police rather than directly giving them food and money.

Conversely, corruption within charities associated with the government has had a profoundly negative impact on the willingness to donate and volunteer. Participants referred to the infamous corruption scandal involving the Red Cross, where an influencer named 郭美美 (Guo Meimei) allegedly received money from Red Cross leaders to finance luxury [18]. Additionally, participants claimed that some of the "volunteering work" organized by communities is, in fact, a form of embezzlement disguised as volunteer activities. They observed that positions for such volunteering work are often reserved for the relatives and friends of leaders, creating discriminatory practices that hinder equal opportunities. Although officials officially state that there is no personal gain from volunteering, participants alleged that money and gifts are still offered.

Homelessness is a sensitive topic. Chinese participants expressed doubts about the authenticity of homeless individuals. Statements such as "99% of the homeless in China are fake/fraud," "They earn more than me," and "They don't want food instead of money" reflect participants' scepticism regarding the identity of the homeless. Many participants believe that homeless individuals possess the ability and opportunities to work but choose begging as a relatively easier activity.

**3.2.4 Reflect through diary and express feelings.** Chinese participants tend to engage in writing self-reflection dairy less frequently and are generally less inclined to express their emotions towards their friends.

3.2.4.1. Writing a self-reflection diary.

*"What kind of decent person writes a diary?" ["Audino", Female, 24, Engineering Researcher] & ["Shuckle", Male, 26, HCI Researcher] & ["Zapdos", Male, 42, Technical Director]*

Being a common practice in the west, writing a self-reflection diary regularly was repeatedly considered "weird" by the Chinese participants across different workshops. Some believe that only those who have something to remorse or something that can evoke a lot of emotions write a diary. People also accuse it of being emotionally exhausting and may evoke negative feelings that they do not know how to deal with afterwards.

3.2.4.2. Expressing love towards people around.

"Telling friends how important they are to me. I think this only happens when someone is drunk, in my experience." ["Bellossom", Female, 28, Athelete]

Different from the direct and explicit expression of emotions and thoughts commonly observed in Western culture, participants argue that Chinese tend to express their feelings in a more subtle and implicit way.

**3.2.5 Family.** Many participants believe that interacting with relatives (especially those who are not that close) is in fact the opposite of relaxation and is emotionally exhausting. Therefore, some argue that because of the traditional values of family, it is a responsibility rather than a spontaneous act to maintain the relationship between family members, which can be stressful and negative towards mental well-being. "Call a relative", "visit a relative" and "plan a trip to visit the hometown" are all ruled out as positive activities.

"Set up a family calendar", which is believed to boost the efficiency of communication among family members for 4.3% of the participants, is also considered stressful by another

19.6%. As sharing the schedules with other family members gives some participants the feeling of violation of their personal boundaries. "List chores and split with family members" is also considered to hold the risk of provoking family conflicts that they would rather avoid.

**3.2.6 Religion.** China, as a secular state, is characterised by a prevailing absence of religious adherence among its population [19]. Consequently, it is unsurprising that 30.4% of the participants expressed reservations regarding the inclusion of the activity of prayer within the social framework.

*"As individuals of Chinese heritage, we abstain from subscribing to the notion of 'Monsters and Demons."'* ["Cyndaquil", Male, 28, Consultant] The participants invoked the historical context of the *"Sweep Away All Monsters and Demons"* campaign during the era of Chairman Mao, during which gods and ghosts were considered antithetical to communist ideology. Moreover, for numerous Chinese individuals, engaging in prayer is regarded as an evasive behaviour, intended to avoid confronting underlying issues.

However, it is important to note that there are participants who derive solace from religious practices, such as prayer and partaking in religious events such as temple visits, to attain inner tranquillity [20].

## 3.3 Work-related controversial activities

The workshops showed that cultural differences significantly influence employees' perspectives on work-related activities. These activities, when compared to non-work-related conflicts, often raise concerns that can potentially deteriorate workplace relationships and even lead to job loss.

### 3.3.1 Supervisor-subordinate relationship

Initially, four activities were included in the Australian App Headgear designed to facilitate communication between employees and their leaders. These activities, including providing creative advice, adjusting work hours, offering flexible working arrangements, and granting pay raises, where there is a strong emphasis on employee autonomy in terms of communication and interaction with management. However, all four activities sparked debates and conflicting comments during our workshops.

*"In China, you definitely can't do this. I will definitely give it a cross. You are digging your own grave."* ["Chikorita", Male, 25, Auditor]

"Discuss rescheduling work hours/pay raise with your leader? Meaning you want to quit your job?" ["Purrloin", Male, 27, Construction Worker]

*"Not only will you be rejected, but your leader will also make things hard for you afterwards."* ["Chikorita", Male, 25, Auditor]

Such remarks were repeated ten times across the twelve workshops, highlighting employees' concerns about the potential risks of job loss associated with these actions. It is crucial to consider and avoid these risky suggestions when designing well-being support apps.

The influence of the strict supervisor-subordinate relationship is also evident in the organisation of work meetings. Among the participants, 37% voted against the activity of "organising walking meetings at work." This resistance stems from the belief that meetings, especially in state-owned enterprises and government agencies, serve not only to communicate work progress but also to consolidate leadership authority.

However, for the idea related to "discussing flexible working hours with leaders" some of our participants expressed support. They argued that after many companies experimented with remote work during the COVID-19 pandemic, employers have recognised the potential benefits of flexible working.

**3.3.2 Inter-employee relationships.** Similarly, dealing with inter-employee relationships can also induce significant stress among many employees. One activity that has garnered criticism is "participating in events with colleagues," which has evoked miserable memories for 41.3% of the participants. An essential aspect of *guanxi*, the Chinese concept of interpersonal relationships, is maintaining harmony between individuals, even in the presence of mutual animosity. *"In many Chinese companies, the prevailing company culture and environment require individuals to wear a metaphorical mask at all times, feigning happiness and friendliness towards everyone while silently harbouring negative sentiments." ["Inteleon", Female, 27, General Manager]* Even engaging in workplace sports societies is deemed to be *"solely about cultivating social connections"*, adding to people's stress levels. Another point of contention against these activities is that they are often organised outside of working hours, further burdening already exhausted employees.

**3.3.3 Working hours.** Long working hours are an inescapable issue when considering the well-being of Chinese employees. In recent years, "996" (working from 9 am to 9 pm, six days a week) has become synonymous with excessive overtime, sometimes being replaced by the extreme "007" (working 12 hours a day, seven days a week). This exploitative practice [21] undoubtedly harms employees' physical and mental well-being [15]. Additionally, some companies foster an inter-employee competition culture, eliminating those who do not display initiative in working overtime. The combination of long working hours and a fiercely competitive working environment exacerbates the detriment to employee well-being.

One participant shared an example of a daily quote received on their enterprise communication platform, stating, *"I love a flower without asking how long it blooms. I love a job, no matter when I can get off work, I just work from early morning to late night." ["Magikarp", Female, 26, Biomedical Researcher]*

## 3.4 Mental health experts

Participants generally displayed hesitancy towards the activity of "making an appointment with a psychologist/counsellor", citing the immaturity of psychiatric and counselling services in China due to their inaccessibility and high costs. Some participants with prior experience with mental health professionals expressed concerns about their expertise, thereby worrying about the potential risks of exacerbating their conditions through misdiagnosis.

## 3.5 Amendments and newly proposed activities

Through the workshops we identified many cultural differences in the set of activities considered positive for wellbeing. Critiques, amendments, and alternatives proposed by participants for 28 controversial activities are summarised in Table 1. High NE scores suggest the potential controversies between Chinese and Australian employees. Low DF scores suggest the existence of controversies among Chinese participants. The thematic analysis around each activity was further critiqued to understand the reasons behind the controversies.

In total, 219 new suggestions were proposed by the participants during the workshops. Most of them can be adopted as supplements to existing activities, leaving 38 additional popular new ideas presented in Table 2.

**Table 1. Amendments to controversial activities.**

| English Name (abbreviated) | NE (%) | DF (%) | Critique | Amendments | Alternatives Suggested by participants |
|---|---|---|---|---|---|
| Water sport | 15.2 | 2.2 | Not accessible | Lower the priority | |
| Go extreme | 21.7 | -4.3 | Not accessible; Risk; Not broadly accepted | Remove | |
| Join community sports team | 23.9 | -10.9 | Not accessible | Lower the priority | |
| Swim before work | 43.5 | -28.3 | Not accessible; Contradict with work time; Physically exhausting | Rephrase to "Swim" | |
| Buy some new fitness gear | 21.7 | -2.2 | Incomprehensible | Rephrase to "Exercise with new fitness gear" | |
| Attend a trial lesson in gym | 4.3 | 0.0 | Consumption trap; Potential scam | Remove | |
| Mow the lawn | 43.5 | -32.6 | Not accessible | Remove | "Plant flowers and vegetables" |
| Start setting aside 50¥ per week to put towards a monthly sporting or cultural event | 30.4 | -23.9 | Incomprehensible; Contradict with Chinese financial habit | Remove | |
| Become a community volunteer | 26.1 | -2.2 | Government responsibility; Corruption mistrust | Remove | "Donate or volunteer to help stray animals" |
| Participate in charitable events | 13 | 6.5 | Government responsibility; Corruption mistrust | Remove | |
| Provide food for homeless | 32.6 | -17.4 | Government responsibility; Potential scam | Remove | "Give money to street performers" |
| Write a diary | 23.9 | -2.2 | Emotionally exhausting; Evoke negative feelings | Rephrase to "Write down happy or interesting moments" | |
| To see the family | 8.7 | 0.0 | Emotionally exhausting | Remove | |
| Visit relatives | 21.7 | -4.3 | Emotionally exhausting | Remove | |
| Call relatives | 26.1 | -4.3 | Emotionally exhausting | Remove | "Get to know relatives' worries and offer help" |
| Start a family calendar | 19.6 | -15.2 | Privacy | Remove | |
| Divide up chores between family members | 13.0 | -6.5 | Provoking family conflicts | Remove | |
| Telling friends how important they are to me | 23.9 | -2.2 | Contradict with Chinese expression | Remove | "Prepare a surprise gift for friends" |
| Sign up to a course or volunteer to meet new people | 13 | 6.5 | Not broadly accepted | Rephrase to "Sign up to a course to meet new people with similar interests" | |
| Pray | 30.4 | -15.2 | Not broadly accepted; Too religious | Remove | "Debate" |
| Offer creative advice to leaders | 19.6 | 8.7 | Contradict with Chinese workplace culture; Risk | Remove | "Help establish rules and policies" |
| Discuss possibility for rescheduling work hours | 17.4 | 2.2 | Contradict with Chinese workplace culture; Risk | Remove | "Rent a flat near to the office" |
| Discuss a pay raise with leaders | 19.6 | -2.2 | Contradict with Chinese workplace culture; Risk | Remove | "Invest" |
| Organise to have walking meetings at work | 37.0 | -26.1 | Contradict with Chinese workplace culture; Not broadly accepted | Remove | |
| Participate in events with colleagues | 41.5 | -17.4 | Emotionally exhausting; Disrupting work life balance | Remove | |
| Join a work sports team | 28.3 | -4.3 | Emotionally exhausting | Remove | |
| Create a family calendar for the fridge and schedule all activities on this | 17.4 | 2.2 | Emotionally exhausting; Evoke negative feelings | Rephrase to "Schedule time for relaxation" | "Remove useless plans" |
| Visit psychologist/counsellor | 19.6 | 4.3 | Risk; Financial burden | Provide reliable and professional medical institution information | "Learn psychology and get to know myself" |

**Table 2. Other popular suggestions.**

| Category | Suggested Activities (organised and translated into English) |
|---|---|
| Recreational | Sing at home or go to a KTV |
| | Go to a musical, play, concert, or Chinese comedy |
| | Cook (with friends and family) |
| | Listen to music (while doing chores) |
| | Go to a bookstore or a library |
| | Play a room escape or murder mystery game |
| | Take a bath, foot bath, or go to the hot spring |
| | Enjoy comic, novel, anime, audio book, and TV drama |
| | Read news |
| | Zone out or take a good sleep |
| | Read history or visit historical sites and cities |
| | Practice calligraphy |
| Helping Others | Be a volunteer teacher in poorer areas |
| Friends & Family | Play video games, go shopping, read books, or learn together with friends |
| | Offer massage, travel, or watch something together with family members |
| | Keep a pet (with the partner) |
| | Remember and celebrate special dates |
| | Send red packets to each other |
| Health | Get health monitoring devices |
| | Do eye relaxation exercises or get the device for it |
| | Try boxing, kung fu, or taichi |
| | Cry or shout out loud |
| | Watch relaxing, cute, or stimulating videos |
| Work & Study | Share experience and skills in internal seminars and blogs |
| | Help others to communicate efficiently |
| | Get inspired by others' great work |
| | Invent new approaches and technologies |
| | Start a self-media |
| | Learn how to boost working efficiency |
| | Work on a personal project with colleagues and friends |
| | Discuss about work with friends |
| Daily Life | Buy a dishwasher |
| | Room decoration |
| | Throw away meaningless or unnecessary stuff |
| | Summarise work/life/outcomes regularly |
| | Follow a healthy routine |
| | Delete unnecessary social apps and friends |
| | Refuse unwanted gatherings |

# 4. Discussion

## 4.1 Principal findings

Behavioural activation therapies have shown to be effective in the west, but the recommended behaviours are highly dependent on the culture of the target audience. Little is known about the behaviours perceived to contribute to wellbeing in China.

The objective of this study is identifying "positive activities" suitable for behavioural activation therapy within the cultural context of China.

Based on the workshop feedback from 12 workshops, we have analysed and found that the previously successful activities proven to work in the Australian population [7] need to be re-adapted for cross-cultural effectiveness among Chinese employees. We have also observed variations within the Chinese employee group towards certain activities. To better comprehend the root causes of these conflicting activities, which include both intergroup differences between Australian and Chinese employees and within-group differences, we plan to conduct further analysis using Hofstede's cultural dimension theory with consideration of China's specific socio-cultural context.

The cultural dimension theory did not guide the design process of the method, which is primarily based on [17]. However, it will be used in this section as a theoretical framework to explore "controversial activities," which was mentioned in the previous section. The specific cultural context of China compared to Western countries will also be taken into the consideration.

**4.1.1 Hierarchical culture with fierce competition.**   According to Hofstede's cultural dimensions theory, the power distance dimension measures the extent to which members of a society accept and expect hierarchical power and authority distribution. The masculinity-femininity dimension, on the other hand, assesses the value placed by a society on achievement, competition versus care, collaboration, and quality of life. In the context of Chinese culture, which exhibits high scores in power distance and masculinity, there is a strong emphasis on respecting authority and advocating obedience towards those in positions of power. Confucianism has historically justified hierarchical social stratification through the dictum: *"The honor bestowed upon the sovereign is on par with that of Heaven, akin to that of a parent. When the people revere Heaven, they are in fact praising their fathers. "* [22]. Moreover, there exists fierce competition to gain power, as individuals from higher social classes enjoy greater privileges and societal moral standing, also referred to as *mianzi* or "face."

As a result of these cultural dynamics, certain Western "positive activities" are deemed unsuitable in the Chinese environment due to cultural differences. For instance, activities like "Offering creative advice to leaders", "Discussing the possibility of rescheduling work hours", or "Negotiating a pay raise with superiors", were effective in the Australian workplace,are considered unrealistic by many participants. A recent workplace culture survey conducted by [23] in China revealed that only 4.4% of employees perceive managers prioritise emphasis on establishing an "employee-oriented" approach. In China, it is expected that junior individuals show absolute respect and obedience towards their senior counterparts in order to maintain their seniors' mianzi. The supervisor-subordinate guanxi, or relationship, within Chinese workplace culture entails positive reciprocal exchanges, but it can also result in impression management targeted towards preserving the superiors' mianzi. This, in turn, discourages employees from expressing dissenting opinions or providing constructive feedback, as they fear the potential loss of face for their bosses. In other words, successful communication between an employee and their leader necessitates not only presenting well-founded proposals but also exhibiting sufficient respect to leave a positive impression, which is equally, if not more, important. In contrast, Western cultures, such as Australia, have lower power distance scores, which means that they prioritise individuality and personal freedom. This cultural emphasis enables employees to provide feedback to their superiors more effectively. Additionally, compared to China, many western countries have more feminine cultures foster a people-oriented work environment where employee well-being is considered as important as the company's profits. Work-related activities that generate controversy are heavily influenced by cultural differences and can raise ethical concerns. In contrast to the comparatively egalitarian and collaborative working relationships in Australia, the rigid supervisor-subordinate dynamic in Chinese workplaces introduces additional complexity to these dynamics. While some activities face

resistance due to concerns about job security, others gain support due to changing attitudes towards flexible work arrangements. Understanding these dynamics is crucial when developing well-being support apps for employees.

It is worth noting that this dynamic is subject to change, particularly with the entrance of "Generation X" into the workplace, as they were born under the "one-child policy" [23]. Younger generations in China may exhibit more individualistic tendencies and display less acceptance of strict workplace hierarchies. Two studies have yielded results that contradict Hofstede's conclusions. Hamid [24] found no evident preference for images depicting authority when analysing representations on Chinese university websites. Similarly, in 2022, Yang conducted a cross-cultural comparison between Chinese and US users of TikTok/Douyin and surprisingly discovered that Chinese participants claimed higher individualism scores and lower power distance scores than their US counterparts [25]. Recent surveys have also revealed a growing trend among the younger generation, particularly millennials and Generation X, who exhibit more negative attitudes towards attending socialising events with colleagues after work [23].They are also increasingly reluctant to discuss current events with colleagues in the workplace [26] Interestingly, an emerging trend was observed amidst the predominantly negative sentiment surrounding the workplace. Several participants demonstrated a proactive attitude by proposing constructive work-related activities, such as "Collaboratively establishing rules and policies", "Sharing experiences and skills through internal seminars and blogs.","discussing flexible working hours with leaders"—participants argued that after the COVID-19 pandemic, Chinese employer also recognised the benefits of flexible working. Flexible working became new normal [27].

Furthermore, the concept of family is also undergoing significant changes with ongoing social transformations [28, 29] While the family has historically been regarded as the foundation of Chinese culture, many young participants express reluctance to visit relatives or maintain regular contact with family members. This aversion could also be attributed to the hierarchical and power distance dynamics that exist within the family unit. The activity of "Creating a family calendar for the fridge and scheduling all activities on it" was considered emotionally draining and likely to evoke negative feelings among many participants. Chinese family gatherings often involve social comparisons among family members and the experience of pressure from senior family members. Additionally, with the emergence of the internet and increasing mobility of the population [30], families now rely less on each other for financial and emotional support, which has led to looser familial connections across generations. We need to be aware of the cross-generation difference when designing and carrying out our intervention.

**4.1.2 High-context culture and group conformity.**   The individualism-collectivism dimension measures the extent to which a culture values personal choice, autonomy, and self-expression compared to adherence to group norms and maintaining group harmony. Interestingly, certain activities related to emotional expression, such as "writing in a diary" and "telling friends how important they are to me," which were well-received by Australian participants, elicited feedback that either categorised them as "emotionally exhausting" or as "contradictory to Chinese expression."

This can be attributed to China's collectivist cultural orientation, where group harmony is highly valued, and the well-being of the collective takes precedence over individual subjective feelings or expressions. Countries with individualistic cultures, such as Australia, score high on individualism, place value on self-expression, autonomy, and direct communication. However, in Chinese culture, placing excessive emphasis on personal emotions or expressing emotions too directly may be perceived as drawing attention to oneself and disrupting the group dynamic, which is considered undesirable. As a result, Chinese individuals may choose to

express their emotions indirectly or in a more subdued manner to avoid standing out or causing discomfort to others. This aligns with Hall's concept of high-context cultures [31], where communication is conveyed through context, nonverbal cues, shared experiences, and implicit assumptions. Expressions are often indirect and implicit, rather than straightforward.

Charitable acts, which was considered a way to bring about positive emotions in the previous study in Australia [7], also received negative feedback from Chinese participants In an Individualistic country, the equality of human rights is highly valued regardless of social status, whereas in a collectivist country like China, characterised by high power distance, group harmony is highly valued, and social status and hierarchy hold great significance. This may lead to negative attitudes towards disadvantaged individuals, who might be viewed as burdens or incapable of contributing to society. Meanwhile, disadvantaged individuals could be perceived as a threat to social stability and group conformity. Furthermore, considering China's unique context, communities (also referred to as residential units) are government-associated entities overseen by subdistrict administrators, where dedicated committee members are paid to serve the community. Consequently, many participants view it as the government's responsibility, rather than that of individuals, to help those in need. This perspective extends to issues related to homelessness, as participants believe that the Chinese government already possesses well-developed policies for addressing homelessness, given the existence of government departments that provide food and shelter for the homeless.

The uncertainty avoidance dimension measures the extent to which a society avoids ambiguity and relies on rules and procedures to mitigate the unpredictability of future events. Visiting a psychologist or counsellor is another activity that generated controversy. One reason for the employees' hesitation to "make an appointment with a psychologist/counsellor" is the underdeveloped clinical mental health system, as mentioned earlier in the results section. Unlike Australia, where the mental health system is highly developed, well-established, and widely accessible, the regulation of the mental health industry in China remains insufficient, despite the gravity of the mental health issues faced by Chinese employees. In September 2017, the Ministry of Human Resources and Social Security of China issued a notice titled "Announcement on the Publication of the National Occupational Qualification Catalogue" (MHRSS [2017] No. 68), which resulted in the discontinuation of the professional qualification examination for "Psychological Counsellor," commonly known as the "Psychological Counsellor Certificate." From a cultural perspective, the low score in uncertainty avoidance partially explains the Chinese clinical mental health system's environment, which exhibits greater tolerance for ambiguity and fewer explicit regulations. The absence of legislative and professional boundaries contributes to a lack of trust in mental health services. Compared to China, Australia has a higher score of uncertainty avoidance, society including the mental health system, operates in an orderly manner under the supervision of laws and regulations.

In addition to the underdeveloped support system, another potential barrier to seeking mental health professional help is the sense of shame associated with the process. This can be attributed to the enduring mental health stigma prevalent in collectivist cultures like China. A study conducted in Tianjin [32] revealed that a significant portion of the Chinese population holds negative attitudes towards individuals with mental illness, particularly within close personal relationships. The study considered cultural influences from Confucianism, including pejorative beliefs about mental illnesses and the significance of *mianzi*, where mental health issues could be considered a disgrace as they violate the social norm. The study also found a negative association between mental health knowledge and public stigma, indicating a lack of proper understanding of mental illness causes, treatments, and prevention in China.

It is noteworthy that many workshop participants proposed alternative solutions for clinical mental health services, such as watching videos of cute pets or kids, having a pet (mentioned

by three different participants), and learning psychology to gain self-awareness. Gender differences were observed in proposing alternative solutions, with many suggestions coming from female participants, especially those with less than three years of work experience.

**4.1.3 Suppressing the present desire for the bigger picture.**   Long-term orientation refers to the extent to which a culture values persistence and future-oriented behaviour over short-term gratification and living in the present. [33] found that culture can influence saving habits. Their study utilised Hofstede's cultural indices and macro data from 48 countries spanning 1990–2013. The results indicated that culture can largely explain individual differences in savings behaviour, with cultural variables being among the most significant determinants. The Australian culture is characterised by a short-term orientation, prioritise immediate gratification and instant outcomes, while China scores high on long-term orientation. This cultural difference could account for the confusion experienced by Chinese participants when they encounter Westerners expressing enthusiasm for "saving $50 per week for sport and art." For Chinese participants, saving money is considered a natural financial hbit rather than something requiring specific motivation to initiate. The prevailing economic stagnation in China has further enhanced the tendency of a significant majority of Chinese participants to prioritise saving money over consumption [34].The indulgence-restraint dimension measures the degree to which a culture values pleasure, desire, and enjoyment versus restraint and self-discipline. Australia has a higher score in indulgence compared to China, leading to a greater openness to enjoying life and a constant pursuit of new experiences, while Chinese culture tends to be more restraint and self-disciplined, thus Chinese and Australian hold different attitudes towards extreme activities. For instance, in the original activity of "Going extreme," participants were encouraged to plan snow trips, go skydiving, bungee jumping, engage in quad-biking, play paintball, climb mountains, or participate in abseiling. In traditional Chinese societies influenced by Confucian values, filial piety holds significant importance as a social discipline. The body is regarded as a gift from parents and should be preserved as an expression of filial piety [35]. Moreover, the principle of "The Way of Harmonious Equilibrium" in Chinese culture emphasises balance and moderation, discouraging participation in extreme sports due to perceived risks and potential harm to health and well-being. On another hand, the accessibility of such activities also need to be taken into consideration. Research conducted by [36], which investigated leisure activities among Chinese individuals, revealed that people in Beijing predominantly engage in daily leisure activities within their local neighbourhoods, primarily due to accessibility and time constraints. Unlike Australians, who have convenient access to swimming pools and beaches, many Chinese individuals reside far from city centres due to high office rents, leading to lengthy daily commutes of up to four hours. This observation aligns with the findings presented in the Annual Report on Leisure Development in China [18] which highlight that residents of 10 major cities in China allocate more time to sports and fitness activities after retirement, when they have fewer work obligations and more leisure time. This explains why "water sports" and "joining sports teams in your area," which were generally considered positive activities in Australia, were not endorsed by many Chinese participants. Participants have suggested more relaxed, easy-to-access, recreational options as alternatives, such as "Dancing to music" and "Planting flowers/vegetables." These suggestions offer milder and easier alternatives for individuals seeking a leisurely and enjoyable experience.

## 4.2 Limitations

The workshops involved 46 Chinese employees. Although the researcher considered gender, age, working area, and educational level distributions, bias may still exist due to the small

sample size. Additionally, most participants, regardless of income levels, were recruited from major cities (provincial capitals). Including employees from minor cities and rural areas in further studies would provide valuable insights. Additionally, due to the pandemic, we were unable to conduct additional face-to-face workshops. While online anonymity may encourage participants to speak more openly, in-person discussions could potentially yield more in-depth conversations.

### 4.3 Conclusions

This study aimed to investigate effective behavioural activation activities specifically designed for Chinese employees, considering the traditional influence of Confucianism and the ongoing socio-cultural transformation of China. The findings revealed the necessity to adapt behavioural activation activities to align with Chinese culture and current circumstances. It is crucial to acknowledge that Chinese employees often face work-related pressure, necessitating careful considerations when connecting work to their personal time, and make sure the potential activity would not invoke additional stress. Given the shifts in generations, it is important to consider the variations within each sub social group. For example, this could include the integration of age-specific features and activities. Drawing upon Hofstede's framework and Chinese culture, several recommendations emerge. Firstly, it is advisable to provide milder activity suggestions that address effective communication with authority figures. Secondly, it is crucial to recognise the significance of indirect expression in Chinese interpersonal relationships and the nuances of mental health regulation in China, offering mental health self-help suggestions and providing education on mental health knowledge can serve as alternative options. Furthermore, lighter and less risky sports and entertainment activities are recommended. Considering the Chinese cultural value of long-term orientation, it is essential for those activities to effectively communicate the lasting benefits of participation to Chinese employees. To ensure successful implementation, these adjustments should be incorporated both before and during the actual intervention for behavioural activation therapy. By tailoring the activities to Chinese cultural norms and considering the pressures faced by Chinese employees, this study seeks to enhance the effectiveness and relevance of behavioural activation interventions within this population.

## Supporting information

**S1 Checklist. STROBE checklist.**
(DOCX)

**S1 File. Activities in the previous Australian study.**
(XLSX)

**S2 File. Anonymised qualitative results.**
(PDF)

**S3 File. Anonymised statistical results.**
(XLSX)

## Author Contributions

**Conceptualization:** Sijin Sun, Celine Mougenot, Rafael A. Calvo.

**Data curation:** Sijin Sun.

**Formal analysis:** Sijin Sun, Yao Xiao, Zheyuan Zhang.

**Investigation:** Sijin Sun.

**Methodology:** Sijin Sun, Rafael A. Calvo.

**Project administration:** Sijin Sun.

**Supervision:** Celine Mougenot, Nick Glozier, Rafael A. Calvo.

**Validation:** Sijin Sun.

**Visualization:** Sijin Sun.

**Writing – original draft:** Sijin Sun, Yao Xiao.

**Writing – review & editing:** Sijin Sun, Yao Xiao, Zheyuan Zhang, Celine Mougenot, Nick Glozier, Rafael A. Calvo.

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
