## [Decision Letter · Decision Letter 0]

18 Apr 2024

PMEN-D-23-00069

Behavioural activation activities for Chinese employees: a workshop

PLOS Mental Health

Dear Dr. Sijin Sun

Thank you for submitting your manuscript to PLOS Mental Health. After careful consideration, we feel that it has merit but does not fully meet PLOS Mental Health’s publication criteria as it currently stands. Therefore, we invite you to submit a revised version of the manuscript that addresses the points raised during the review process.

Specifically, all the reviewers indicate that the methodology used should be made more clearer in the manuscript. The description of the protocol should be provided to a greater detail to enable the readers appreciate the process and the findings. 

We look forward to receiving your revised manuscript.

Kind regards,

Martin Mabunda Baluku, Ph.D.

Academic Editor

PLOS Mental Health

Journal Requirements:

1. Please provide separate figure files in .tif or .eps format and remove the embedded figures from the manuscript file.

https://journals.plos.org/mentalhealth/s/figures 

https://journals.plos.org/mentalhealth/s/figures#loc-file-requirements 

2. In the online submission form, you indicated that "For concerns of participants' privacy, the data in this study are not published in public database. Anonymised data could be provided upon requests.". 

3. Uploaded as supplementary information.

Additional Editor Comments (if provided):

Reviewers' comments:

Reviewer's Responses to Questions

**Comments to the Author**

1. Does this manuscript meet PLOS Mental Health’s publication criteria? Is the manuscript technically sound, and do the data support the conclusions? The manuscript must describe methodologically and ethically rigorous research with conclusions that are appropriately drawn based on the data presented.

Reviewer #1: No

Reviewer #2: Partly

Reviewer #3: Yes

Reviewer #4: Partly

2. Has the statistical analysis been performed appropriately and rigorously?

Reviewer #1: I don't know

Reviewer #2: No

Reviewer #3: Yes

Reviewer #4: I don't know

3. Have the authors made all data underlying the findings in their manuscript fully available (please refer to the Data Availability Statement at the start of the manuscript PDF file)?

Reviewer #1: Yes

Reviewer #2: No

Reviewer #3: Yes

Reviewer #4: Yes

4. Is the manuscript presented in an intelligible fashion and written in standard English?

Reviewer #1: Yes

Reviewer #2: Yes

Reviewer #3: Yes

Reviewer #4: Yes

5. Review Comments to the Author

Reviewer #1: Overall: Please include page and line numbers. It is also important to justify (distribute text evenly between margins)

Title “Behavioural activation activities for Chinese employees: a workshop” why not behavioral activation therapy for Chinese employees to reflect the exact intervention under consideration. The aim of the study is also not well reflected in the title.

Abstract: Include information of method of data collection and analysis.

Background: Background is well written, although it relies most on the comparison between Australia and China, yet the title does not portray the study as a comparative study

2.1 Objectives and overview: since the study approach is mentioned under this section, please provide a justification for the choice of approach. Information about participants recruitment should be moved to 2.2.2 ‘participants’

It is also unclear to me if this was a mixed methods study. From the text, it is difficult to understand but looking at figure 1, it looks like one. Offer clarity on this, and if it’s a mixed methods study, mention the exact type of MM used and a justification for the choice.

2.2 Workshop: Adapting digital BAT is a bit different from adapting a face-to-face BAT. Looks like the authors were interested in digital BAT but this does not come out well in the background.

2.2.1 ‘Participants’. Give additional details about the recruitment process. How did the potential participants get to know about this study? How was the consent process handled?

#Participant’s demographics should be moved to the results section and put in a table

2.2.2 Materials and settings: Information presented under these sections is basically about ‘procedure’. The contact and the title contradict.

#Can the authors include a table summarizing the workshop contents? Clearly indicating the number of workshops and the duration the workshops lasted? What was the criteria of assigning participants to different workshops?

2.2.3 Protocol: It remains unclear to me whether all participants were attending one online workshop at a time or they were in separate workshops. In case of the latter, what was the consideration for assigning participants to the separate workshops? Who facilitated the discussions? Where these recorded? Earlier, the authors mention that the workshops varied from 1.5 -3 hours, what caused the variation in time? did this time cover the overall presentation, presentation of perspectives and discussing possible modifications?

2.2.1. Data collection: Experience of living abroad first appears under this section, any justification?

# I have completely failed to understand the method of data collection used to collect participants proposals on new activities.

#Was there a data collection guide used during the workshops? If Yes, can sample questions be included here and the guide provided as supplementary material?

2.3.2 Data analysis: I have many questions about this section, particularly the qualitative part. Earlier alone the authors indicated that the study was guided by Hofstede’s framework, I have not seen that guide data analysis. Second, if the study was guided by a framework, how can data be analyzed inductively? What was the role of the framework? The data analysis process is also not detailed and robust, it is difficult to understand how data was delt with. More details are needed.

Were the workshops recorded? And then data transcribed? If so, who translated the data to English? Were participants involved in the translation to avoid misrepresentation?

# The robustness of qualitative analysis is found in the details of how the data was handled. This information is not provided anywhere. Who analyzed the data? What was their level of qualifications?

# Given that there was a lot of qualitative data collected, a statement of trustworthiness needs to be included. Secondly how was social desirability delt with?

# Since the end goal was identifying elements of the BAT for cultural and contextual adaptation, using an adaptation framework would have helped the study. The study seems to have been targeted at assessing social validity of BAT and if this was the case, did participants have sufficient knowledge about the BAT to be able to provide useful feedback beyond liking and not liking? In the composition of participants where mental health experts (who blend both professional knowledge and understanding of the context) involved?

3. Results: Better to start this section with reporting participants demographics

#Controversial activities” was this the main aim of the study? You seem to be giving it too much importance.

3.1. “These” Controversial Activities” were observed at two distinct layers. Firstly, the results conveyed culture-specific messages that diverged from the corresponding findings in Australia”

Was this a comparative study? If yes, then it should be reflected in the title and background.

#If the study was guided by Hofstede’s framework, it should then guide presentation of results and the discussion.

3.1.1 Sports: Participant voices are not represented in this section of results. Adding quotes would have improved trustworthiness of the results

# The arrangement of the manuscript is a bit confusing. Looks like the discussion starts right away from presentation of results. I suggest the results section is only left for results.

3.3 Mental health experts “Unlike Australia, the mental health system is highly developed, well-established, and widely accessible, despite the gravity of the mental health issue for Chinese employee, the regulation of the mental health industry in China remains insufficient” Is this a statement made by participants? Given that this is a results section, could the authors just leave it to results?

# This statement could be revised to bring out the intended meaning.

# citing the immaturity of psychiatric and counselling services in China due to their inaccessibility and high costs. Some participants with prior experience with mental health professionals expressed concerns about their expertise, thereby worrying about the potential risks of exacerbating their conditions through misdiagnosis” Is this a finding from the current study? I suggest the authors resist from speaking beyond the results.

Table 3: The second category in Table 3 is misleading “English Name (abbreviated, what exactly are the authors communicating?

#Table 2 and 3 have different formats, can the authors use one format throughout the manuscript?

3.5 Summary of findings: I recommend it is removed because it is a repetition of the introduction part of the results.

4.1 Discussion. The objective of the study listed under the discussion differs from the objective set earlier.

# Under 4.1 authors talk about the intention to conduct further analysis using Hofstede’s cultural dimensions and immediately start presenting results following the dimensions in 4.1.1. when was the analysis mentioned above done to be able to guide the following discussion?

#If the authors wanted Hofstede’s dimensions to guide the discussion, they should have used the dimensions to guide data collection and presentation of results. There are so many things mixed up.

# The discussion needs to be improved to clearly reflect the study results.

4.2 Limitations. Authors list small sample size as a potential cause of bias. If the study methods were clear, this could have been handled well. 46 participants in a qualitative study cannot be considered to be a small size, however, since they study also talks about quantitative data somewhere, this is where the confusion started from.

Reviewer #2: Thank you for the opportunity to read and review this article.

I have a few comments below.

Introduction

1 What is a "more positive behaviours"? It seems that there are less positive behaviours.

2. The authors have made an effort to justify the topic and emphasise the relevance of the study. I believe that the reader should have a clearer definition of the terms.

3. I don't see the need for table 1.

Methods

1. The title 2.1. is unclear. It says that it is about the aim of the study, but nothing is said about the aim. In fact, the objective has already been presented, so the title is not appropriate.

2. What was the criterion for selecting the participants? Where in China are the participants from?

3. In point 2.2.2. it's not clear what the materials are.

4. Figure 2 is not clear.

5. What statistical analysis was carried out?

Results

1. The abbreviations in table 2 are not understood (e.g. NE, DF).

2. What statistical treatment was carried out?

3. It is not clear why the results observed are often compared with data from other studies with Australians.

Reviewer #3: The study presents a thorough examination of the workshop's design, materials, and protocol, offering a transparent portrayal of the methodology. Additionally, the researchers conducted an exhaustive analysis of the data, encompassing both quantitative assessment of participants' engagement with activities and qualitative examination of discussions. However, certain crucial aspects necessitate further elaboration:

1. Inclusion and Exclusion Criteria: The methods section should provide explicit details regarding the criteria used for participant selection, including any exclusion criteria employed. The authors should clearly specify whether individuals with pre-existing mental health conditions, such as depression or anxiety, were excluded from the study.

2. Participant Demographics: The authors need to provide more detailed information on demographic data of partcipants such as age, gender,etc. Also, the participants' work variables, such as type of occupation and working hours. Additionally, the relationship between participant demographics, such as gender, and the study's aims should be discussed.

3. Response Rate and Decline Rate: Given the online nature of the study, it is imperative for the authors to disclose the response rate, indicating the proportion of invitations accepted, as well as the decline rate. These statistics offer insights into the sample representativeness and potential biases.

4. Dropout Rate: Throughout the duration of the research, tracking the dropout rate is essential. This provides an understanding of participant attrition and its potential impact on the study's outcomes.

5. Limitations: The authors should acknowledge the limitations of the online workshop format, which may have limited the depth of interactions and discussions compared to in-person sessions.

Reviewer #4: Thank you to the authors for studying an underreported topic in behavioral activation activities within a Chinese cohort; highlighting regional differences from previous research. Though the topic is well founded, there are a few recommendations to help amplify the quality of the manuscript.

Methods: Though attached separately; a sentence specifying accordance with STROBE guidelines should be included in the methods of the manuscript. Additionally, further elaboration regarding the participants of the study if possible, by specifying whether any participants were present at >1 workshop; and whether any compensation was given would help improve transparency. Lastly, please specify whether the 201 activities included from the Headgear app were the total number present on the application, or a subset. If a subset of activities were selected for the study, the reason for such selection would be needed.

Results: Please specify demographics data that was mentioned in the methods in the results, as that would allow for further consideration of potential skew impacting the results (a younger population within the study may be inclined to select certain activities). Additionally, this study utilizes the results section to compare data with data from the previous study on an Australian cohort. Such comparisons should be restricted to the discussion section of the manuscript.

Discussion: The discussion brings good points to compare the pre-existing dynamics within China and it's impact on activities preference. However, at times such points are not supported with references, such as the discussion on workplace culture. If possible, please add supporting references from literature to further solidify the points made by the discussion.

Overall, this paper presents a novel idea with the implementation of a good study design to collect qualitative data. Additionally, the rationale for the implementation is well founded and expanded upon in the introduction, with the discussion making good points regarding the disparities noted in contrast to previous studies.

6. PLOS authors have the option to publish the peer review history of their article (what does this mean?). If published, this will include your full peer review and any attached files.

**Do you want your identity to be public for this peer review?** For information about this choice, including consent withdrawal, please see our Privacy Policy.

Reviewer #1: No

Reviewer #2: No

Reviewer #3: No

Reviewer #4: No

---

## [Decision Letter · Decision Letter 1]

1 Oct 2024

PMEN-D-23-00069R1

Behavioural activation activities for employees in the Chinese culture: a workshop

PLOS Mental Health

Dear Dr. Sun,

Thank you for submitting your manuscript to PLOS Mental Health. After careful consideration, we feel that it has merit but does not fully meet PLOS Mental Health’s publication criteria as it currently stands. Therefore, we invite you to submit a revised version of the manuscript that addresses the points raised during the review process.

We look forward to receiving your revised manuscript.

Kind regards,

Ansar Abbas

Academic Editor

PLOS Mental Health

Journal Requirements:

Additional Editor Comments (if provided):

Reviewers' comments:

Reviewer's Responses to Questions

**Comments to the Author**

1. If the authors have adequately addressed your comments raised in a previous round of review and you feel that this manuscript is now acceptable for publication, you may indicate that here to bypass the “Comments to the Author” section, enter your conflict of interest statement in the “Confidential to Editor” section, and submit your "Accept" recommendation.

Reviewer #1: All comments have been addressed

2. Does this manuscript meet PLOS Mental Health’s publication criteria? Is the manuscript technically sound, and do the data support the conclusions? The manuscript must describe methodologically and ethically rigorous research with conclusions that are appropriately drawn based on the data presented.

Reviewer #1: Yes

3. Has the statistical analysis been performed appropriately and rigorously?

Reviewer #1: Yes

4. Have the authors made all data underlying the findings in their manuscript fully available (please refer to the Data Availability Statement at the start of the manuscript PDF file)?

Reviewer #1: Yes

5. Is the manuscript presented in an intelligible fashion and written in standard English?

Reviewer #1: Yes

6. Review Comments to the Author

Reviewer #1: Authors have adequately addressed all the suggestions but need to address just a few issues before the manuscript is accepted for publication (see attachment)

7. PLOS authors have the option to publish the peer review history of their article (what does this mean?). If published, this will include your full peer review and any attached files.

**Do you want your identity to be public for this peer review?** For information about this choice, including consent withdrawal, please see our Privacy Policy.

Reviewer #1: No

---

## [Decision Letter · Decision Letter 2]

5 Nov 2024

Behavioural activation activities for employees in the Chinese culture: a workshop

PMEN-D-23-00069R2

Dear Miss Sun,

We are pleased to inform you that your manuscript 'Behavioural activation activities for employees in the Chinese culture: a workshop' has been provisionally accepted for publication in PLOS Mental Health.

Best regards,

Ansar Abbas

Academic Editor

PLOS Mental Health

Reviewer Comments (if any, and for reference):

Reviewer's Responses to Questions

**Comments to the Author**

1. If the authors have adequately addressed your comments raised in a previous round of review and you feel that this manuscript is now acceptable for publication, you may indicate that here to bypass the “Comments to the Author” section, enter your conflict of interest statement in the “Confidential to Editor” section, and submit your "Accept" recommendation.

Reviewer #1: All comments have been addressed

2. Does this manuscript meet PLOS Mental Health’s publication criteria? Is the manuscript technically sound, and do the data support the conclusions? The manuscript must describe methodologically and ethically rigorous research with conclusions that are appropriately drawn based on the data presented.

Reviewer #1: Yes

3. Has the statistical analysis been performed appropriately and rigorously?

Reviewer #1: N/A

4. Have the authors made all data underlying the findings in their manuscript fully available (please refer to the Data Availability Statement at the start of the manuscript PDF file)?

Reviewer #1: Yes

5. Is the manuscript presented in an intelligible fashion and written in standard English?

Reviewer #1: Yes

6. Review Comments to the Author

Reviewer #1: Authors have addressed all my comments

7. PLOS authors have the option to publish the peer review history of their article (what does this mean?). If published, this will include your full peer review and any attached files.

**Do you want your identity to be public for this peer review?** For information about this choice, including consent withdrawal, please see our Privacy Policy.

Reviewer #1: **Yes: **Khamisi Musanje
